**Data Availability Statement:** The data for these analyses were licensed by Magellan Health, Inc.,

# Interferon gamma release assay tests are associated with persistence and completion of latent tuberculosis infection treatment in the United States: Evidence from commercial insurance data

**Erica L. Stockbridge**[1,2]*, **Abiah D. Loethen**[1], **Esther Annan**[3], **Thaddeus L. Miller**[2]

1 Department of Advanced Health Analytics and Solutions, Magellan Health, Inc., Scottsdale, Arizona, United States of America, 2 Department of Health Behavior & Health Systems, School of Public Health, University of North Texas Health Science Center, Fort Worth, Texas, United States of America, 3 Department of Biostatistics and Epidemiology, School of Public Health, University of North Texas Health Science Center, Fort Worth, Texas, United States of America

* estockbridge@magellanhealth.com

## Abstract

### Background

Risk-targeted testing and treatment of latent tuberculosis infection (LTBI) is a critical component of the United States' (US) tuberculosis (TB) elimination strategy, but relatively low treatment completion rates remain a challenge. Both treatment persistence and completion may be facilitated by diagnosing LTBI using interferon gamma release assays (IGRA) rather than tuberculin skin tests (TST).

### Methods

We used a national sample of administrative claims data to explore associations diagnostic test choice (TST, IGRA, TST with subsequent IGRA) and treatment persistence and completion in persons initiating a daily dose isoniazid LTBI treatment regimen in the US private healthcare sector between July 2011 and March 2014. Associations were analyzed with a generalized ordered logit model (completion) and a negative binomial regression model (persistence).

### Results

Of 662 persons initiating treatment, 327 (49.4%) completed at least the 6-month regimen and 173 (26.1%) completed the 9-month regimen; 129 (19.5%) persisted in treatment one month or less. Six-month completion was least likely in persons receiving a TST (42.2%) relative to persons receiving an IGRA (55.0%) or TST then IGRA (67.2%; p = 0.001). Those receiving an IGRA or a TST followed by an IGRA had higher odds of completion compared to those receiving a TST (aOR = 1.59 and 2.50; p = 0.017 and 0.001, respectively). Receiving an IGRA or a TST and subsequent IGRA was associated with increased treatment persistence relative to TST (aIRR = 1.14 and 1.25; p = 0.027 and 0.009, respectively).

from Optum, a commercial data provider in the US. As such, the authors cannot provide access to the data themselves. However, other researchers could access the same data by purchase through Optum. Interested individuals may visit https://www.optum.com/business/solutions/data-analytics/data/real-world-data-analytics-a-cpl/claims-data.html for more information on accessing Optum Clinformatics data. We confirm that no authors had special privileges to access data from Optum, and that other researchers would be able to access the data in the same manner as the authors.

**Funding:** ADL is employed at Magellan Health, Inc, a commercial managed care organization. Additionally, Magellan has a contractual arrangement with the University of North Texas Health Science Center, and ELS works with Magellan through this contractual relationship. Magellan provided salary support and access to the data, but Magellan did not have any additional role in the study design, analysis, decision to publish, or preparation of the manuscript. No other authors have financial disclosures to report.

**Competing interests:** No authors have competing interests to report. ADL is employed at Magellan Health, Inc, a commercial managed care organization. Additionally, Magellan has a contractual arrangement with the University of North Texas Health Science Center, and ELS works with Magellan through this contractual relationship. These affiliations do not represent competing interests and do not alter the authors' adherence to PLOS ONE policies on sharing data and materials.

## Conclusions

IGRA use is significantly associated with both higher levels of LTBI treatment completion and treatment persistence. These differences are apparent both when IGRAs alone were administered and when IGRAs were administered subsequent to a TST. Our results suggest that IGRAs contribute to more effective LTBI treatment and consequently individual and population protections against TB.

## Introduction

Risk-targeted testing and treatment of latent tuberculosis infection (LTBI) must become more widespread to markedly accelerate the United States' (US) currently slow progress towards tuberculosis (TB) elimination [1–3]. The importance of LTBI treatment in TB prevention is underscored by the over 80% of active TB cases in the US stemming from latent *Mycobacterium tuberculosis* infections often acquired long before progression to disease [4]. Had these latent infections been proactively identified and treated to completion, up to 90% of these LTBI-related TB cases could have been prevented [5].

A major barrier to mitigating active TB risk by treating LTBI is the often low rate of treatment completion. Up to 61% of those initiating LTBI treatment do not complete their medication regimens, effectively blunting the potential benefits afforded by the substantial public and other investments required to identify and engage treatment-eligible individuals with LTBI [6–13]. Improving treatment completion rates is an attractive and potentially efficient means to improve individual and population protections against TB and advance domestic elimination efforts.

Diagnostic confidence may influence completion rates, and it is plausible that there are important differences in treatment completion following diagnosis via interferon gamma release assays (IGRA) versus tuberculin skin tests (TST). Although TSTs have been in widespread use for over a century [14] they may yield false-positive test results in persons who have had bacille Calmette-Guerin (BCG) vaccines [15,16]. There is widespread use of BCG vaccination in countries other than the US [17] and LTBI prevalence is highest in non-US-born persons [18,19], so it is plausible that confidence in TST as a diagnostic presents a barrier in such key high-risk populations [8,20,21]. Conversely, IGRA results are unaffected by BCG vaccination [15,16] and patients and/or providers may have greater confidence in IGRA test results [8,20,21]. Further, even non-BCG vaccinated individuals and/or their providers may have greater trust in the results of the newer IGRAs relative to the older TSTs because of societal beliefs that "new is better" [22]. Perceptions of medication necessity and effectiveness are associated with medication adherence [23,24]; consequently, the diagnostic confidence afforded by IGRAs may prompt more patients to persist with and complete treatment relative to those diagnosed using TSTs.

Available reports of association between LTBI testing method and treatment completion provide inconclusive and mixed evidence, four finding no significant difference in treatment completion rates by diagnostic method and two reporting an association between higher completion rates and diagnosis using IGRA [20,21,25–28]. All but one of these studies focused on safety net or other public programs [25] and may not be generalizable to the private healthcare sector setting. Further, serial testing is not uncommon and may affect completion rates, but the single private system completion study categorized patients based only on the first test received and did not evaluate serial testing [25]. This is particularly concerning given that clinical practice guidelines state that when persons at low risk for LTBI are tested and the results are positive, a second test is appropriate [15]. Finally, none of the available studies examined test type relative to persistence of treatment in persons not completing treatment, though it is

known that even incomplete isoniazid (INH) and possibly other regimens reduce progression from LTBI to TB disease [29].

Up to 13 million people in the US have LTBI [18,19] and there is increasing recognition that expanding the role of private physician's offices and clinics as partner to public health agencies may be critical to addressing such a large reservoir of TB risk [1,30,31]. Research to support a clearer and more complete understanding of best practices for LTBI testing relative to treatment persistence and completion beyond the public health system is an important and necessary step in the evolving management of US TB risk. This study seeks to provide such evidence, and to fill knowledge gaps around serial testing and partial treatment. We hypothesized that private sector healthcare data will reveal that, relative to TSTs, there are significant associations between IGRAs and both a higher likelihood of treatment completion and greater treatment persistence. We also hypothesized that these associations will hold true when IGRAs alone are administered as well as when an IGRA is administered subsequent to a TST.

## Methods

This project was reviewed and approved as exempt category research by the North Texas Regional Institutional Review Board at the University of North Texas Health Science Center.

### Data source

De-identified medical and pharmacy claims data from the Optum Clinformatics Data Mart were used for the current study. This data source contains claims for roughly 30.6 million people in the US who have commercial insurance [32]. The analytical sample represents claims data describing the healthcare services and medications received by 4 million randomly selected people between 0 and 64 years of age who were continuously enrolled in a commercial health insurance plan between 2011 and 2013, inclusive. Claims data were available for services occurring and prescriptions filled from January 2011 through either the end date of each individual's insurance coverage or March 31, 2015, whichever came first. The geographic distribution of these 4 million persons approximated that of the 2010 US population based on census divisions.

We used a previously published algorithm [9] to identify persons initiating a 6 to 9 month daily dose LTBI treatment regimen of isoniazid between July 2011 and March 2014. Such regimens have been the most commonly used treatments for LTBI [10,33–35]. In accordance with the algorithm logic, we required that the data be available to determine if LTBI treatment was completed. We also required that a Current Procedural Terminology (CPT) code for a TST or IGRA [14] be present on a claim within the 6 months prior to treatment initiation, and we required that data describing the individuals' counties of residence be available.

### Measures

**Outcome variables.**   We examined two outcomes of interest. The first was completion of treatment for LTBI with a daily-dose isoniazid regimen. Because our data do not indicate if the 6 or 9 month regimen was prescribed, we grouped treatments into three mutually exclusive ordinal categories based on the number of doses dispensed, consistent with past research [25]. These categories were 1) non-completion (<180 doses in 9 months), 2) 6-month regimen completion but not 9-month completion (180 to 269 doses within 9 months), 3) 9 month regimen completion (> = 270 doses within 12 months) [25,33].

The second outcome of interest was treatment persistence as measured by a count of the number of months of isoniazid dispensed to each individual initiating isoniazid treatment. While most prescription fills occurred in monthly intervals (i.e., 30 daily doses of medication received on a roughly monthly basis over many months), there were exceptions to this norm.

When an individual's dose counts did not correspond with an interval of 30, the dose count was rounded down to the nearest month count (e.g., someone receiving 65 doses of isoniazid was counted as receiving two months of medication, someone receiving 100 doses was counted as receiving three months, etc.). Additionally, the month count variable was top-coded to 9 months; anyone receiving >9 months was counted as receiving 9 months. No more than 270 doses (9 months) of daily isoniazid is recommended for LTBI treatment [35]; few people received more than that. Examining increasing levels of isoniazid regimen completion and increasing isoniazid treatment persistence are both of interest given that, as the duration of isoniazid treatment increases, the likelihood of progression to active TB disease decreases [29].

**Primary explanatory variable.** The primary explanatory variable was a categorical variable that detailed the pattern of LTBI testing received within the 6 months prior to isoniazid treatment initiation. Specifically, TST and IGRA testing was identified based on CPT codes, with code 86580 identifying TSTs and codes 86480 and 86481 identifying IGRAs [14]. No otherwise eligible persons had data containing CPT codes 0010T or 86585 so they were not considered in our analysis. Testing patterns were categorized based on which tests were received and, if multiple tests were received, the order of the tests. We created four categories: 1) One TST with no subsequent TSTs or IGRAs, 2) One IGRA with no subsequent TSTs or IGRAs, 3) An initial TST followed by one IGRA, but no additional TSTs or IGRAs, 4) Some other pattern of multiple tests. We also conducted a sensitivity analysis to determine if or how the analysis results would change if category 3 also included individuals who received an initial TST followed by at least one IGRA as well as additional TSTs and/or IGRAs; in the categorization described above these persons were included in category 4.

**Explanatory covariates.** To control for potential confounders in the relationship between treatment completion or persistence and testing method, explanatory covariates were included in analyses. Many of these variables were selected based on their associations with treatment completion as identified in past literature [25]. Covariates included sex, age, census region, and urban-rural classification [36]. We used the percentage of households living under the federal poverty level in an individual's county of residence as a proxy of household income. We also included health insurance type (preferred provider organization [PPO], point of service [POS], or health maintenance organization [HMO]), isoniazid prescription size when initially dispensed, and state TB rate [37]. Prevalence of non-US born individuals within the county of residence was used as a proxy for foreign birth [38]. Clinical risk was represented with a simple count of selected clinical risk factors for each individual (i.e., 0, 1, or >1) based on whether individuals' data included evidence of diabetes, tobacco use, a history or late effects of TB, contact with or exposure to TB, HIV, and/or immune-suppressive medication use [25,39]. Details regarding the logic used to create these variables are available elsewhere [25].

## Statistical analyses

We calculated the proportion of individuals in each level of isoniazid treatment completion (i.e., non-complete, 6-month completion, 9-month completion) and the proportion of individuals at each level of isoniazid treatment persistence (i.e., 1 month through 9 months dispensed). We then examined unadjusted associations between the explanatory variables and the two outcome variables using Kruskal-Wallis tests or Spearman's correlations. We examined the adjusted association between treatment completion and the LTBI testing pattern using a multivariable generalized ordered logit model. Variables meeting the parallel-lines assumption of the ordered logit model were constrained to have equal effects, so for those variables the odds ratios were the same for non-completion versus 6-month completion and <9-month completion versus 9-month completion. Variables violating the assumption were

**Table 1. Distribution of the number of months of isoniazid dispensed to each person initiating latent tuberculosis infection treatment (i.e., treatment persistence) and corresponding treatment completion categorizations (n = 662).**

| # Months of Isoniazid within 1 Year Post-Initiation | Completed At Least a 6 Month Regimen? | Completed 9 Month Regimen? | Count of Persons | Percent | Cumulative Percent |
|---|---|---|---|---|---|
| ≥9 | Yes | Yes | 173 | 26.1% | 26.1% |
| 8 | Yes | No | 59 | 8.9% | 35.0% |
| 7 | Yes | No | 41 | 6.2% | 41.2% |
| 7* | No | No | 2 | 0.3% | 41.5% |
| 6 | Yes | No | 54 | 8.2% | 49.7% |
| 6* | No | No | 9 | 1.4% | 51.1% |
| 5 | No | No | 45 | 6.8% | 57.9% |
| 4 | No | No | 42 | 6.3% | 64.2% |
| 3 | No | No | 59 | 8.9% | 73.1% |
| 2 | No | No | 49 | 7.4% | 80.5% |
| 1 | No | No | 129 | 19.5% | 100.0% |

* Treatment completion of a 6-month isoniazid regimen requires the receipt of at least 180 doses in the 9-month period after treatment initiation [33]. People represented by these rows received at least 180 doses, but not within the 9-month post-initiation period. Additional details describing the dispensation patterns for these persons is available in S1 Table.

not constrained and, as a result, different completion category comparisons will have different odds ratios [40]. We examined the adjusted association between treatment persistence and the LTBI testing pattern using a multivariable negative binomial regression model.

We conducted the statistical testing with Stata 14.2 [StataCorp; College Station, TX]. Significance was tested at $p < .05$, and all statistical testing was two-sided.

## Results

We identified 1074 people who initiated isoniazid treatment for LTBI and had sufficient data to determine if treatment was completed. Two (0.2%) were excluded from analysis due to missing county-related variables and 410 (38.2%) were excluded because their data contained no CPT codes for IGRAs or TSTs in the 6 months prior to treatment initiation, resulting in an analytic sample of 662 persons. Of the 662 persons initiating treatment, 327 (49.4%) completed at least the 6-month regimen, 173 (26.1%) completed the 9-month regimen, and 335 (50.6%) completed neither. More than one-third of all persons initiating treatment (237/662; 35.8%) discontinued treatment within the first three months and 19.5% (129/662) discontinued treatment within the first month. The distribution of the number of months of isoniazid dispensed to each individual and their treatment completion categorizations are detailed in Table 1.

The characteristics of persons in our sample are detailed in Table 2. Slightly over half were tested with a TST and no other tests in the 6 months preceding treatment initiation (n = 339;

**Table 2. Characteristics of 662 persons initiating isoniazid treatment for latent tuberculosis infection.**

| | n | % or Mean of Total | 95% Confidence Interval | |
|---|---|---|---|---|
| | | | Lower | Upper |
| Pre-Treatment Testing Pattern | | | | |
| One TST | 339 | 51.21% | 47.39% | 55.01% |
| One IGRA | 191 | 28.85% | 25.52% | 32.43% |
| One TST followed by one IGRA | 58 | 8.76% | 6.83% | 11.17% |
| Other pattern of multiple TSTs and/or IGRAs | 74 | 11.18% | 8.99% | 13.82% |
| Sex | | | | |

*(Continued)*

**Table 2.** (Continued)

|  | n | % or Mean of Total | 95% Confidence Interval | |
|---|---|---|---|---|
|  |  |  | Lower | Upper |
| Female | 358 | 54.08% | 50.26% | 57.85% |
| Male | 304 | 45.92% | 42.15% | 49.74% |
| Age Group |  |  |  |  |
| 0–14 | 88 | 13.29% | 10.91% | 16.11% |
| 15–29 | 182 | 27.49% | 24.22% | 31.03% |
| 30–44 | 184 | 27.79% | 24.51% | 31.34% |
| 45–64 | 208 | 31.42% | 27.99% | 35.07% |
| Census Region |  |  |  |  |
| Northeast | 210 | 31.72% | 28.28% | 35.38% |
| Midwest | 101 | 15.26% | 12.71% | 18.21% |
| South | 94 | 14.20% | 11.74% | 17.08% |
| West | 257 | 38.82% | 35.17% | 42.60% |
| Patient Location |  |  |  |  |
| Large Central Metro County (Urban) | 313 | 47.28% | 43.49% | 51.10% |
| Large Fringe Metro County (Suburban) | 253 | 38.22% | 34.58% | 41.99% |
| Any Smaller County | 96 | 14.50% | 12.01% | 17.40% |
| % of Households Under FPL in County |  |  |  |  |
| <15% | 374 | 56.50% | 52.68% | 60.24% |
| ≥15% | 288 | 43.50% | 39.76% | 47.32% |
| Insurance Type |  |  |  |  |
| HMO | 98 | 14.80% | 12.29% | 17.73% |
| POS | 473 | 71.45% | 67.88% | 74.77% |
| PPO | 91 | 13.75% | 11.32% | 16.59% |
| Supply of Isoniazid Received on Date of 1st Fill |  |  |  |  |
| < 2 month supply | 611 | 92.30% | 90.00% | 94.10% |
| ≥2 month supply | 51 | 7.70% | 5.90% | 10.00% |
| Period Regimen Started |  |  |  |  |
| 2011 Q3-4 | 142 | 21.45% | 18.48% | 24.75% |
| 2012 Q1-4 | 261 | 39.43% | 35.76% | 43.21% |
| 2013 Q1-4 | 227 | 34.29% | 30.76% | 38.00% |
| 2014 Q1 | 32 | 4.83% | 3.44% | 6.76% |
| State TB Rate/100,000 | - | 3.89 | 3.78 | 4.00 |
| % Foreign Born in County | - | 20.95 | 20.00 | 21.91 |
| Count of Clinical Risk Factors* |  |  |  |  |
| None | 407 | 61.48% | 57.70% | 65.12% |
| 1 | 192 | 29.00% | 25.66% | 32.59% |
| 2 or more | 63 | 9.52% | 7.50% | 12.01% |

* Clinical risk factors included diabetes, tobacco use, a history or late effects of TB, contact with or exposure to TB, HIV, and immune-suppressive medication use.

Abbreviations

HIV: Human immunodeficiency virus

HMO: Health maintenance organization

IGRA: Interferon gamma release assay

POS: Place of service

PPO: Preferred provider organization

Q1: Quarter 1 of the calendar year

Q1-4: Quarters 1–4 of the calendar year (full year)

Q3-4: Quarters 3 and 4 of the calendar year

TB: Tuberculosis

TST: Tuberculin skin test

51.2%), while nearly a third received a single IGRA and no other tests (n = 191; 28.9%). Roughly 20% of our sample received multiple TSTs and/or IGRAs prior to LTBI treatment initiation; 8.8% received one TST and one subsequent IGRA (n = 58) and 11.2% received some other pattern of multiple tests (n = 74) (Table 2). These other patterns included receiving more than two tests, receiving the same kind of test twice, or receiving both types of tests on the same date.

Table 3 provides information about the unadjusted associations between each of the explanatory variables and both treatment completion and the number of months of isoniazid dispensed. Table 4 contains the results of both the multivariable generalized ordered logit model examining the adjusted association between treatment completion and LTBI test and the multivariable negative binomial regression model examining the adjusted association between the number of months of isoniazid dispensed and LTBI test. In all analyses (Tables 3 and 4), the type of LTBI test was significantly associated with both LTBI treatment completion and treatment persistence (i.e., the number of months of isoniazid dispensed). Specifically, in unadjusted analyses (Table 3) treatment completion was least likely in persons receiving only a TST, with only 42.2% completing either a 6-month or 9-month regimen. This relatively low completion rate is in contrast with the 55.0% rate of completion in those receiving only an IGRA, the 67.2% completion rate in those receiving a TST followed by an IGRA, and the 54.0% completion rate in those with some other pattern of multiple tests (p = 0.001; Table 3). A similar pattern is observed when comparing the relatively low median number of months of isoniazid dispensed in those receiving only a TST (median = 5) to the medians in those with only an IGRA (median = 6), those with a TST followed by an IGRA (median = 8), and those with some other pattern of multiple tests (median = 6; p<0.001).

In adjusted analyses (Table 4), persons receiving only an IGRA or those receiving a TST followed by an IGRA had higher odds of treatment completion, relative to those receiving only a TST (Adjusted Odds Ratio [aOR] = 1.59 and 2.50, respectively; p = 0.017 and 0.001, respectively; Table 4). Similarly, receiving only an IGRA and receiving a TST and a subsequent IGRA were each associated with a greater number of months of isoniazid being dispensed, relative to the receipt of a TST only (Adjusted Incidence Rate Ratio [aIRR] = 1.14 and 1.25, respectively; p = 0.027 and 0.009, respectively; Table 4). Conversely, the odds of treatment completion did not differ significantly between persons with some other pattern of multiple tests and those receiving only a TST (aOR = 1.55; p = 0.091; Table 4), and there was no significant difference in the number of months of isoniazid dispensed to those receiving a TST only versus those with some other pattern of multiple tests (aIRR = 1.10; p = 0.230; Table 4). Our sensitivity analysis indicated that our findings are robust to variations in the categorization of persons with an initial TST followed by at least one IGRA as well as additional TSTs and/or IGRAs; see S1 File.

Our adjusted analyses indicated that a number of other explanatory variables were significantly associated with both treatment completion and months of isoniazid dispensed (Table 4). Persons in the 30–44 year age group were less likely than those in the 0–14 year age group to complete treatment (aOR = 0.51; p = 0.010) or or persist in treatment (aIRR = 0.081; p = 0.010). Persons in counties with a higher proportion of households living under the federal poverty level were less likely to complete treatment (aOR = 0. 54; p = 0.001) or persist in treatment (aIRR = 0.084; p = 0.004) relative to those in more affluent counties. Having a preferred provider organization (PPO) insurance plan was associated with a greater likelihood of treatment completion (aOR = 2.89; p<0.001) and greater treatment persistence (aIRR = 1.29; p = 0.004) relative to having a health maintenance organization (HMO) plan. Receiving more than a 1-month supply of isoniazid the first time the prescription was filled was positively associated with both achieving at least 9 months of treatment completion (aOR = 3.01; p<0.001) and greater treatment persistence (aIRR = 1.21; p<0.027). Relative to having no clinical risk

**Table 3. Unadjusted associations between sample characteristics and latent tuberculosis infection (LTBI) treatment.** Three treatment measures are examined: 1) level of isoniazid treatment completion, 2) completion of 6 months of isoniazid, and 3) the number of months of dispensed isoniazid (n = 662).

| | Level of LTBI Treatment Completion | | | | | | | | | | ≥6 Months of LTBI Treatment Completed | | | | 3) Number of Months of Isoniazid Dispensed Within 1 Year of Initiation | | | |
|---|---|---|---|---|---|---|---|---|---|---|---|---|---|---|---|---|---|---|
| | Neither Regimen Completed: <6 Months (Row % or Mean) | | | ≥6 but <9 Months Complete (Row % or Mean) | | | ≥9 Months Complete (Row % or Mean) | | | p-value: 3 Completion Levels | ≥6 Months Complete (Row % or Mean) | | | p-value: <6 vs. ≥6 Months Complete | Median # Months Dispensed | 95% Confidence Interval | | p-value |
| | % or Mean | 95% Confidence Interval Lower | Upper | % or Mean | 95% Confidence Interval Lower | Upper | % or Mean | 95% Confidence Interval Lower | Upper | | % or Mean | 95% Confidence Interval Lower | Upper | | | Lower | Upper | |
| **Pre-Treatment Testing Pattern** | | | | | | | | | | | | | | | | | | |
| One TST | 57.82% | 52.47% | 62.98% | 22.12% | 18.01% | 26.87% | 20.06% | 16.12% | 24.68% | | 42.18% | 37.02% | 47.53% | | 5 | 4 | 5 | |
| One IGRA | 45.03% | 38.09% | 52.17% | 25.13% | 19.47% | 31.79% | 29.84% | 23.76% | 36.74% | | 54.97% | 47.83% | 61.91% | | 6 | 5 | 7 | |
| One TST followed by one IGRA | 32.76% | 21.87% | 45.88% | 25.86% | 16.15% | 38.72% | 41.38% | 29.39% | 54.48% | | 67.24% | 54.12% | 78.13% | | 8 | 6 | 9 | |
| Other pattern of multiple TSTs and/or IGRAs | 45.95% | 34.89% | 57.41% | 21.62% | 13.63% | 32.53% | 32.43% | 22.71% | 43.95% | <0.001 | 54.05% | 42.59% | 65.11% | 0.001 | 6 | 5 | 7 | <0.001 |
| **Sex** | | | | | | | | | | | | | | | | | | |
| Female | 52.23% | 47.04% | 57.38% | 22.91% | 18.83% | 27.56% | 24.86% | 20.64% | 29.62% | | 47.77% | 42.62% | 52.96% | | 5 | 5 | 6 | |
| Male | 48.68% | 43.09% | 54.32% | 23.68% | 19.23% | 28.81% | 27.63% | 22.88% | 32.95% | 0.334 | 51.32% | 45.68% | 56.91% | 0.363 | 6 | 5 | 6 | 0.213 |
| **Age Group** | | | | | | | | | | | | | | | | | | |
| 0–14 | 42.05% | 32.14% | 52.64% | 25.00% | 17.01% | 35.15% | 32.95% | 23.90% | 43.48% | | 57.95% | 47.36% | 67.86% | | 7 | 6 | 8 | |
| 15–29 | 52.20% | 44.91% | 59.39% | 24.73% | 18.98% | 31.54% | 23.08% | 17.50% | 29.78% | | 47.80% | 40.61% | 55.09% | | 5.5 | 4 | 6 | |
| 30–44 | 55.98% | 48.70% | 63.01% | 22.83% | 17.31% | 29.48% | 21.20% | 15.86% | 27.73% | | 44.02% | 36.99% | 51.30% | | 5 | 4 | 6 | |
| 45–64 | 48.08% | 41.34% | 54.89% | 21.63% | 16.54% | 27.78% | 30.29% | 24.40% | 36.90% | 0.064 | 51.92% | 45.11% | 58.66% | 0.144 | 6 | 5 | 7 | 0.032 |
| **Census Region** | | | | | | | | | | | | | | | | | | |
| Northeast | 52.38% | 45.60% | 59.08% | 19.05% | 14.27% | 24.95% | 28.57% | 22.85% | 35.08% | | 47.62% | 40.92% | 54.40% | | 5 | 4 | 6 | |
| Midwest | 48.51% | 38.88% | 58.26% | 26.73% | 18.97% | 36.25% | 24.75% | 17.27% | 34.14% | | 51.49% | 41.74% | 61.12% | | 6 | 4 | 6 | |
| South | 57.45% | 47.21% | 67.08% | 20.21% | 13.24% | 29.61% | 22.34% | 15.00% | 31.93% | | 42.55% | 32.92% | 52.79% | | 5 | 3 | 6 | |
| West | 47.47% | 41.41% | 53.61% | 26.46% | 21.41% | 32.21% | 26.07% | 21.05% | 31.80% | 0.549 | 52.53% | 46.39% | 58.59% | 0.361 | 6 | 5 | 7 | 0.320 |
| **Patient Location** | | | | | | | | | | | | | | | | | | |
| Large Central Metro County (Urban) | 48.88% | 43.36% | 54.43% | 24.60% | 20.13% | 29.69% | 26.52% | 21.91% | 31.70% | | 51.12% | 45.57% | 56.64% | | 6 | 5 | 6 | |

*(Continued)*

**Table 3.** (Continued)

| | Level of LTBI Treatment Completion | | | | | | | | | | ≥6 Months of LTBI Treatment Completed | | | | 3) Number of Months of Isoniazid Dispensed Within 1 Year of Initiation | | | |
| | Neither Regimen Completed: <6 Months (Row % or Mean) | | | ≥6 but <9 Months Complete (Row % or Mean) | | | ≥9 Months Complete (Row % or Mean) | | | p-value: 3 Completion Levels | ≥6 Months Complete (Row % or Mean) | | | p-value: <6 vs. ≥6 Months Complete | Median # Months Dispensed | 95% Confidence Interval | | p-value |
| | % or Mean | 95% Confidence Interval | | % or Mean | 95% Confidence Interval | | % or Mean | 95% Confidence Interval | | | % or Mean | 95% Confidence Interval | | | | | | |
| | | Lower | Upper | | Lower | Upper | | Lower | Upper | | | Lower | Upper | | | | Lower | Upper | |
|---|---|---|---|---|---|---|---|---|---|---|---|---|---|---|---|---|---|---|
| Large Fringe Metro County (Suburban) | 52.96% | 46.78% | 59.06% | 20.16% | 15.65% | 25.58% | 26.88% | 21.76% | 32.70% | | 47.04% | 40.94% | 53.22% | | 5 | 4 | 6 | |
| Any Smaller County | 50.00% | 40.06% | 59.94% | 27.08% | 19.10% | 36.89% | 22.92% | 15.55% | 32.44% | 0.797 | 50.00% | 40.06% | 59.94% | 0.623 | 6 | 4 | 6 | 0.570 |
| % of Households Under FPL in County | | | | | | | | | | | | | | | | | | |
| <15% | 46.79% | 41.77% | 51.88% | 23.80% | 19.74% | 28.39% | 29.41% | 25.00% | 34.25% | | 53.21% | 48.12% | 58.23% | | 6 | 5 | 7 | |
| ≥15% | 55.56% | 49.75% | 61.22% | 22.57% | 18.09% | 27.78% | 21.88% | 17.46% | 27.04% | 0.014 | 44.44% | 38.78% | 50.25% | 0.026 | 5 | 4 | 6 | 0.023 |
| Insurance Type | | | | | | | | | | | | | | | | | | |
| HMO | 63.27% | 53.25% | 72.26% | 18.37% | 11.85% | 27.35% | 18.37% | 11.85% | 27.35% | | 36.73% | 27.74% | 46.75% | | 4 | 3 | 5 | |
| POS | 49.68% | 45.18% | 54.19% | 24.95% | 21.24% | 29.06% | 25.37% | 21.64% | 29.50% | | 50.32% | 45.81% | 54.82% | | 6 | 5 | 6 | |
| PPO | 41.76% | 32.03% | 52.17% | 19.78% | 12.79% | 29.31% | 38.46% | 29.00% | 48.89% | 0.004 | 58.24% | 47.83% | 67.97% | 0.010 | 6 | 5 | 8 | 0.006 |
| Supply of Isoniazid Received on Date of 1st Fill | | | | | | | | | | | | | | | | | | |
| <2 month supply | 51.39% | 47.42% | 55.35% | 24.22% | 20.98% | 27.79% | 24.39% | 21.14% | 27.96% | | 48.61% | 44.65% | 52.58% | | 6 | 5 | 6 | |
| ≥2 month supply | 41.18% | 28.48% | 55.17% | 11.76% | 5.33% | 23.99% | 47.06% | 33.76% | 60.79% | 0.015 | 58.82% | 44.83% | 71.52% | 0.161 | 6 | 4 | 9 | 0.004 |
| Period Regimen Started | | | | | | | | | | | | | | | | | | |
| 2011 Q3-4 | 55.63% | 47.34% | 63.63% | 26.06% | 19.47% | 33.93% | 18.31% | 12.75% | 25.58% | | 44.37% | 36.37% | 52.66% | | 5 | 4 | 6 | |
| 2012 Q1-4 | 51.34% | 45.26% | 57.38% | 19.92% | 15.50% | 25.23% | 28.74% | 23.55% | 34.54% | | 48.66% | 42.62% | 54.74% | | 5 | 5 | 6 | |
| 2013 Q1-4 | 48.02% | 41.56% | 54.54% | 25.11% | 19.88% | 31.18% | 26.87% | 21.49% | 33.04% | | 51.98% | 45.46% | 58.44% | | 6 | 5 | 6 | |
| 2014 Q1 | 40.63% | 25.02% | 58.38% | 25.00% | 12.86% | 42.94% | 34.38% | 19.95% | 52.40% | 0.181 | 59.38% | 41.62% | 74.98% | 0.338 | 6 | 2 | 9 | 0.637 |
| State TB Rate per 100,000 | 3.93 | 3.78 | 4.08 | 3.91 | 3.67 | 4.15 | 3.81 | 3.59 | 4.03 | 0.451 | 3.86 | 3.70 | 4.02 | 0.560 | N/A | - | - | 0.807 |

(Continued)

**Table 3.** (Continued)

| | Level of LTBI Treatment Completion | | | | | | | | | | ≥6 Months of LTBI Treatment Completed | | | | 3) Number of Months of Isoniazid Dispensed Within 1 Year of Initiation | | | |
|---|---|---|---|---|---|---|---|---|---|---|---|---|---|---|---|---|---|---|
| | Neither Regimen Completed: <6 Months (Row % or Mean) | | | ≥6 but <9 Months Complete (Row % or Mean) | | | ≥9 Months Complete (Row % or Mean) | | | p-value: 3 Completion Levels | ≥6 Months Complete (Row % or Mean) | | | p-value: <6 vs. ≥6 Months Complete | Median # Months Dispensed | 95% Confidence Interval | | p-value |
| | % or Mean | 95% Confidence Interval | | % or Mean | 95% Confidence Interval | | % or Mean | 95% Confidence Interval | | | % or Mean | 95% Confidence Interval | | | | | | |
| | | Lower | Upper | | Lower | Upper | | Lower | Upper | | | Lower | Upper | | | Lower | Upper | |
| % Foreign Born in County | 20.61 | 19.33 | 21.89 | 20.76 | 18.66 | 22.86 | 21.78 | 19.84 | 23.72 | 0.516 | 21.30 | 19.88 | 22.73 | 0.667 | N/A | - | - | 0.269 |
| Count of Clinical Risk Factors* | | | | | | | | | | | | | | | | | | |
| None | 54.55% | 49.67% | 59.34% | 22.11% | 18.33% | 26.42% | 23.34% | 19.47% | 27.71% | | 45.45% | 40.66% | 50.33% | | 5 | 5 | 6 | |
| 1 | 44.27% | 37.37% | 51.40% | 26.56% | 20.77% | 33.29% | 29.17% | 23.15% | 36.01% | | 55.73% | 48.60% | 62.63% | | 6 | 5 | 7 | |
| 2 or more | 44.44% | 32.63% | 56.92% | 20.63% | 12.31% | 32.50% | 34.92% | 24.13% | 47.52% | 0.029 | 55.56% | 43.08% | 67.37% | 0.038 | 6 | 3 | 8 | 0.026 |

* Clinical risk factors included diabetes, tobacco use, a history or late effects of TB, contact with or exposure to TB, HIV, and immune-suppressive medication use.

Abbreviations

HIV: Human immunodeficiency virus

HMO: Health maintenance organization

IGRA: Interferon gamma release assay

N/A: Not applicable

POS: Place of service

PPO: Preferred provider organization

Q1: Quarter 1 of the calendar year

Q1-4: Quarters 1–4 of the calendar year (full year)

Q3-4: Quarters 3 and 4 of the calendar year

TB: Tuberculosis

TST: Tuberculin skin test

**Table 4. Adjusted associations between sample characteristics and two outcomes: 1) level of latent tuberculosis infection treatment completion with isoniazid, and 2) treatment persistence (i.e., the number of months of isoniazid received in the one year after initiation of latent tuberculosis infection treatment) (n = 662).**

| | Outcome: Level of latent tuberculosis infection treatment completion with isoniazid | | | Outcome: Isoniazid treatment persistence | | | |
|---|---|---|---|---|---|---|---|
| | Adjusted Odds Ratio (aOR) | 95% Confidence Interval of aOR | | p-value | Adjusted Incidence Rate Ratio (aIRR) | 95% Confidence Interval of aIRR | | p-value |
| | | Lower | Upper | | | Lower | Upper | |
| Pre-Initiation Testing Pattern | | | | | | | | |
| One TST | 1.00 (base) | | | | 1.00 (base) | | | |
| One IGRA | 1.59 | 1.09 | 2.33 | 0.017 | 1.14 | 1.02 | 1.29 | 0.027 |
| One TST followed by one IGRA | 2.50 | 1.46 | 4.29 | 0.001 | 1.25 | 1.06 | 1.47 | 0.009 |
| Other pattern of multiple TSTs and/or IGRAs | 1.55 | 0.93 | 2.56 | 0.091 | 1.10 | 0.94 | 1.28 | 0.230 |
| Sex | | | | | | | | |
| Female | 1.00 (base) | | | | 1.00 (base) | | | |
| Male | 1.09 | 0.81 | 1.48 | 0.569 | 1.05 | 0.96 | 1.16 | 0.271 |
| Age Group | | | | | | | | |
| 0–14 | 1.00 (base) | | | | 1.00 (base) | | | |
| 15–29 | 0.63 | 0.38 | 1.05 | 0.076 | 0.88 | 0.76 | 1.04 | 0.126 |
| 30–44 | 0.51 | 0.31 | 0.85 | 0.010 | 0.81 | 0.70 | 0.95 | 0.010 |
| 45–64 | 0.64 | 0.37 | 1.09 | 0.098 | 0.88 | 0.75 | 1.04 | 0.128 |
| Census Region | | | | | | | | |
| Northeast | 1.00 (base) | | | | 1.00 (base) | | | |
| Midwest | 0.87 | 0.48 | 1.57 | 0.640 | 1.01 | 0.84 | 1.22 | 0.917 |
| South | 0.86 | 0.50 | 1.49 | 0.598 | 0.95 | 0.81 | 1.13 | 0.571 |
| West | 1.14 | 0.71 | 1.81 | 0.592 | 1.05 | 0.91 | 1.21 | 0.536 |
| Patient Location | | | | | | | | |
| Large Central Metro County | 1.00 (base) | | | | 1.00 (base) | | | |
| Large Fringe Metro County | 0.77 | 0.49 | 1.23 | 0.275 | 0.91 | 0.79 | 1.05 | 0.214 |
| Any Smaller County | 0.87 | 0.49 | 1.55 | 0.643 | 0.99 | 0.83 | 1.18 | 0.898 |
| % of Households Under FPL in County | | | | | | | | |
| <15% | 1.00 (base) | | | | 1.00 (base) | | | |
| ≥15% | 0.54 | 0.37 | 0.79 | 0.001 | 0.84 | 0.75 | 0.95 | 0.004 |
| Insurance Type | | | | | | | | |
| HMO | 1.00 (base) | | | | 1.00 (base) | | | |
| POS | 1.56 | 0.94 | 2.58 | 0.087 | 1.09 | 0.94 | 1.27 | 0.260 |
| PPO | 2.89 | 1.60 | 5.25 | <0.001 | 1.29 | 1.09 | 1.54 | 0.004 |
| Year Regimen Started | | | | | | | | |
| 2011 Q3-4 | 1.00 (base) | | | | 1.00 (base) | | | |
| 2012 Q1-4 | 1.14 | 0.75 | 1.72 | 0.550 | 0.99 | 0.87 | 1.12 | 0.848 |
| 2013 Q1-4 | 1.22 | 0.80 | 1.86 | 0.362 | 1.00 | 0.88 | 1.14 | 0.973 |
| 2014 Q1 | 1.98 | 0.94 | 4.15 | 0.071 | 1.04 | 0.83 | 1.32 | 0.721 |
| State TB Rate | 0.84 | 0.72 | 0.99 | 0.040 | 0.97 | 0.92 | 1.02 | 0.173 |

*(Continued)*

**Table 4.** (Continued)

| | Outcome: Level of latent tuberculosis infection treatment completion with isoniazid | | | Outcome: Isoniazid treatment persistence | | | |
|---|---|---|---|---|---|---|---|
| | Adjusted Odds Ratio (aOR) | 95% Confidence Interval of aOR | | p-value | Adjusted Incidence Rate Ratio (aIRR) | 95% Confidence Interval of aIRR | | p-value |
| | | Lower | Upper | | | Lower | Upper | |
| % Foreign Born in County | 1.01 | 0.99 | 1.03 | 0.195 | 1.00 | 1.00 | 1.01 | 0.152 |
| Count of Clinical Risk Factors* | | | | | | | | |
| None | 1.00 (base) | | | | 1.00 (base) | | | |
| 1 | 1.46 | 1.02 | 2.08 | 0.036 | 1.15 | 1.03 | 1.28 | 0.013 |
| 2 or more | 1.55 | 0.87 | 2.76 | 0.135 | 1.09 | 0.92 | 1.31 | 0.321 |
| Days Supply of Isoniazid Received on Date of 1st Fill | | | | | | | | |
| | Neither regimen completed vs. 6 months completed | | | | Overall | | | |
| < 2 month supply | 1.00 (base) | | | | 1.00 (base) | | | |
| > = 2 month supply | 1.55 | 0.84 | 2.86 | 0.161 | 1.21 | 1.02 | 1.43 | 0.027 |
| | <9 months completed vs. > = 9 months completed | | | | | | | |
| < 2 month supply | 1.00 (base) | | | | | | | |
| > = 2 month supply | 3.01 | 1.63 | 5.55 | <0.001 | | | | |

* Clinical risk factors included diabetes, tobacco use, a history or late effects of TB, contact with or exposure to TB, HIV, and immune-suppressive medication use.

Abbreviations

HIV: Human immunodeficiency virus

HMO: Health maintenance organization

IGRA: Interferon gamma release assay

POS: Place of service

PPO: Preferred provider organization

Q1: Quarter 1 of the calendar year

Q1-4: Quarters 1–4 of the calendar year (full year)

Q3-4: Quarters 3 and 4 of the calendar year

TB: Tuberculosis

TST: Tuberculin skin test

factors, having one risk factor was associated with higher odds of treatment completion (aOR = 1.46; p = 0.036) and greater treatment persistence (aIRR = 1.15; p = 0.013). Further, the state TB rate was inversely associated with likelihood of treatment completion (aOR = 0.84; p = 0.040), although the state TB rate was not significantly associated with treatment persistence (aIRR = 0.97; p = 0.173). Additional details, including confidence intervals and information about non-significant associations, are available in Table 4.

## Discussion

We found that, relative to TSTs, IGRAs were significantly associated with both higher levels of treatment completion and greater treatment persistence. These differences were apparent both

when IGRAs alone were administered and when IGRAs were administered subsequent to a TST (Tables 2 through 4). Our results add to known advantages for LTBI screening via IGRA relative to TSTs. Compared to IGRAs, TSTs are more likely to yield false-positive results when patients have been BCG vaccinated [16], they are more likely to yield false-negative results in immune-suppressed patients [41,42], and diagnosing LTBI with TSTs can be problematic, especially when patients do not return to have TST results interpreted [27,34]. While we are unable to make causal statements based on our data, our findings suggest that IGRAs may also promote LTBI treatment persistence and completion.

Past studies examining the differences in treatment completion for patients receiving IGRAs or TSTs have had varied results, with many finding no difference and only a couple identifying significant differences [20,21,25–28]. However, unlike almost all prior studies, we examined data from the private healthcare sector. Thus, relative to the prior studies conducted on patients receiving treatment at select health departments or in a community-based programs led by health departments [20,21,26–28], our data likely included testing and treatment rendered by providers with a variety of specialties practicing in locations across the US. Relative to public health clinics and providers, these providers likely had a much lower volume of, and potentially less experience or lower comfort level with, TB-related testing or LTBI treatment. While one prior study did examine treatment completion in the private healthcare sector, it examined only the first test administered to each patient. Specifically, it combined persons with a TST only (our reference group) with persons who received an IGRA subsequent to a TST (our third group) [25]; as a result, differences in treatment completion like those that we observed would have been masked.

Our use of private healthcare sector data is also important because there is growing interest in increasing the private sector's role in targeted LTBI testing and treatment [1]. Local public health departments have long been the providers of most TB-related care, including LTBI testing and treatment [30,31,43]. However, public health departments and agencies do not have the capacity or resources to provide LTBI-related services on the scale that is needed to achieve marked progress towards TB elimination [30,31]. Conversely, that capacity exists within the private sector healthcare system, so there is a need to engage providers within that system in conducting targeted LTBI testing and treatment activities. Additionally the US Preventive Services Task Force (USPSTF) has recognized health benefits afforded by LTBI screening in high-risk populations. The USPSTF recommends that such screening be conducted and has provided guidance to primary care physicians to facilitate the appropriate provision of this preventive healthcare service [44]. Consequently, in accordance with Affordable Care Act requirements, most private health insurance plans now provide LTBI screening services to high-risk patients with no out-of-pocket costs to these patients [45]. We are likely to see increased LTBI testing and treatment in the private sector as providers increasingly begin to follow these recommendations and patients begin to access these insurance-covered preventive services.

Even with these facilitating influences low LTBI treatment completion rates remain a barrier to effective TB risk mitigation. While treatment completion rates in the private healthcare sector fall within the range found in public health settings [9], completion rates are often low regardless of setting [6–13]. Further, while we found that IGRAs were associated with higher levels of treatment completion relative to TSTs, we observed that many tested with IGRAs still did not complete treatment; only 55% of those initiating treatment after testing with an IGRA alone and 67% testing with an IGRA subsequent to a TST completed six months or more of isoniazid treatment. Thus, even if IGRAs increase the likelihood of treatment completion, a combination of approaches is likely needed to bring LTBI treatment completion rates closer to optimal levels. Previous research conducted in public health settings suggests that appointment reminders [12], convenient clinic hours [12], and social interventions [46] (e.g., education,

coaching, peer counseling) are associated with improved treatment completion, although many of these interventions have not been found to be consistently effective [8,46]. A notable exception is that short-course LTBI treatment regimens are typically associated with higher treatment completion rates relative to longer six-month or nine-month regimens of isoniazid [7,10,11,13,34,47]; one review found that shorter regimens are associated with a roughly 20% greater treatment completion relative to longer regimens [6]. Given these findings, future research is needed to examine the associations between LTBI treatment completion and test type on patients taking either four months of daily rifampin or three months of once-weekly isoniazid plus rifapentine, the two shorter course regimens currently recommended by the CDC and the National Tuberculosis Controllers Association [34,35].

Additionally, as LTBI treatment becomes more common in the private sector healthcare setting, strategies to improve treatment completion in that environment must be developed. Medication adherence challenges are not unique to LTBI care; patient medication non-adherence is common for many conditions treated in the private sector. We found that a large percentage of patients discontinue LTBI treatment after filling their first prescription; over one-third of patients who did not complete treatment only filled one prescription for isoniazid. This is similar to the patterns seen with previous studies of LTBI treatment in public health settings [47,48] as well as private sector healthcare treatment with antidepressants [49], antipsychotics [50], statins [51], diabetes medications [52], osteoporosis medications [53], and medications prescribed post-cardiac surgery [54]. Thus, interventions that address early treatment discontinuation are critical. A recent systematic review of randomized controlled trials evaluating interventions to improve treatment adherence for a variety of conditions treated in the private sector suggested that complex strategies involving multiple components appeared to be most impactful [55]. This is worrisome given the resources needed to implement such programs. Effective interventions often involved early and ongoing patient-centered support from professionals (e.g., community pharmacists) who provided intense education, counseling, and/or treatment support, and at times these professional supports were supplemented with supports from family or peers [55]. It is especially concerning that these intensive interventions typically yielded only relatively small improvements in treatment adherence [55].

Given the challenges associated with treatment adherence, we must remember that LTBI treatment completion is only one step in the LTBI cascade of care [6,56], and there are opportunities to improve retention throughout this cascade. LTBI treatment is only initiated at the end of a relatively long process. First, persons must be identified as being at risk for LTBI and/or active TB disease. They then must be tested with a TST or IGRA, including a second visit with a provider for a TST to be read if TSTs are used. If the TST or IGRA is positive a medical evaluation including a symptom assessment, physical exam, and chest radiograph must occur for active TB to be ruled out. Then, persons diagnosed with LTBI must receive a recommendation from their provider to initiate treatment and they must agree to initiate the treatment. Finally, those initiating treatment must complete treatment [6,56,57]. Patients may be lost to care throughout this care continuum, and most drop out of the process before they reach the point of treatment initiation–currently it is estimated that only 30.7% of patients with LTBI who were intended for screening actually start treatment [6]. Increasing patient retention throughout this spectrum can yield important efficiencies in terms of outcome for effort, with increasingly large sunk costs invested as a patient moves closer to completion. Preserving the value of this investment as that patient nears the end of the lengthy LTBI care cascade is disproportionately impactful, suggesting that relatively large investments in patient retention as they near completion may be cost efficient. Our results suggest that the use of IGRAs may be part of an effective strategy to preserve these sunk costs by improving LTBI treatment completion and thus forward TB elimination efforts.

While our results provide important insights into LTBI treatment completion, our study does have limitations. While it may be that IGRAs increase diagnostic confidence for providers and/or patients relative to TSTs, we cannot know this with certainty; our data did not allow us to attribute causality. It could be that providers who chose to use IGRAs were more skilled or knowledgeable than those who used TSTs and this knowledge variation is what drove the observed differences in treatment completion, or some other factor might have been at play. We were unable to explore the role of provider specialty or provider and patient knowledge as this information was not available within our data source; there are opportunities for future research that explores these issues.

While the percent of foreign-born persons in a patients' county was used as proxy of nativity, our data also did not contain information about patients' country or birth or BCG vaccination status. Given the likely importance of BCG vaccination in patient and provider concerns about TST accuracy [8,20,21], future studies are needed to determine if BCG vaccination modifies or underpins the observed association between test type and treatment completion. Additionally, two different types of IGRAs were available during the period of study: QuantiFERON-TB Gold In-Tube (QFT-GIT) and T-SPOT. Due to the relative rarity of private sector T-SPOT use during the period of our study [14], these two types of IGRAs were combined for analysis; complicating this is the supplanting of the QFT-GIT in US markets with a newer version, QuantiFERON-TB Gold Plus (QFT-Plus). Still, it is likely that the IGRA effect on treatment persistence and completion is driven more by suspicion of TST than a more direct appreciation for the diagnostic qualities of one or another IGRA product, and it is unlikely that these limitations compromise our findings. Future studies might examine differences between available IGRA tests; in time such research could likely be conducted using administrative claims data because private sector providers' use of these tests has been increasing [14]. Also, as mentioned previously, our study examined treatment completion and duration of isoniazid regimens, so future research is needed to examine associations between completion and duration of newer short-course LTBI treatment regimens and test type. Data limitations left us unable to determine if LTBI treatment was initiated after a false-positive test or whether there are associations between false-positive test results and treatment completion. Still, LTBI is seldom diagnosed or treatment started based solely on test result, and while it is plausible this gap could affect the magnitude of our findings it is unlikely to challenge our conclusions. Additionally, data limitations preclude the direct identification of test results, with a positive TST or IGRA inferred from treatment initiation. This is an important limitation, and it is likely that test choice plays a role in much more than treatment persistence and completion after but also influences very important pre-initiation behavior and decision making such as treatment offer and patient acceptance.

Other limitations of our study are due to the nature of administrative data. Our data enabled us to identify prescriptions that had been filled, but we were unable to determine if the medications had been ingested. Diagnoses and services are typically accurately reflected within claims data and such data provide unique insights into health service use [58], but we cannot be sure whether diagnoses are definitive or presumptive. Further, diagnoses and services that are not coded on a claim are not reflected in the data. Some individuals who initiated LTBI treatment were excluded from analyses because the data did not indicate if they received IGRAs, TSTs, or both. It may be that these persons were receiving LTBI-related medical care from both public and private providers; our data did not provide insights into care delivered across these settings. These types of limitations are typical of studies that use claims data [9,14,25,58–61]; future studies using electronic medical records might provide additional insights.

## Conclusions

Relative to TSTs, IGRAs were significantly associated with both higher levels of LTBI treatment completion and greater treatment persistence. These differences were apparent both when IGRAs alone were administered and when IGRAs were administered subsequent to a TST. Additionally, our study provides insights into LTBI treatment rendered in the private sector healthcare setting, which is likely to be much more heavily represented in US LTBI-related care in the coming years. Low LTBI treatment completion rates have long been a barrier to effective TB risk mitigation, so our finding that IGRAs are associated with persistence and completion is important to private and public sector healthcare providers and public health agencies. Beyond more commonly described advantages of IGRA over TST such as accuracy when testing immune-suppressed and BCG-vaccinated patients and testing in a single visit, our findings suggest that, for many patients, IGRAs appear to be an effective and potentially cost efficient means to promote treatment adherence among the vast but reluctant at-risk US population with LTBI.

## Supporting information

**S1 Table. Prescription fill patterns in persons receiving > = 6 months of isoniazid in a 12 month period who were not categorized as having completed the 6 month regimen.** (PDF)

**S1 File. Sensitivity test examining the impact of varying the categorizations of the pre-treatment testing pattern.** (PDF)

## Acknowledgments

The authors gratefully acknowledge the US Centers for Disease Control and Prevention's Division of Tuberculosis Elimination and its Tuberculosis Epidemiologic Studies Consortium (Atlanta, GA, USA) for its continued guidance and valuable intellectual contributions. Additionally, the research reported in this publication was developed in collaboration with Magellan Health, Inc. (Scottsdale, AZ, USA). We thank Magellan for their invaluable contributions to this work.

The findings and conclusions in this report are those of the authors and do not necessarily represent the official position of the United States Centers for Disease Control and Prevention (CDC) or Magellan Health, Inc. Mention of company names or products does not imply endorsement by the CDC or Magellan.

## Author Contributions

**Conceptualization:** Erica L. Stockbridge, Thaddeus L. Miller.

**Data curation:** Erica L. Stockbridge.

**Formal analysis:** Erica L. Stockbridge.

**Investigation:** Erica L. Stockbridge, Abiah D. Loethen.

**Methodology:** Erica L. Stockbridge, Thaddeus L. Miller.

**Project administration:** Erica L. Stockbridge, Abiah D. Loethen.

**Resources:** Abiah D. Loethen.

**Supervision:** Abiah D. Loethen, Thaddeus L. Miller.

**Validation:** Esther Annan.

**Writing – original draft:** Erica L. Stockbridge.

**Writing – review & editing:** Erica L. Stockbridge, Abiah D. Loethen, Esther Annan, Thaddeus L. Miller.

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
