## [Decision Letter · Decision Letter 0]

24 Sep 2020

PONE-D-20-22812

Interferon gamma release assay tests are associated with persistence and completion of latent tuberculosis infection treatment: Evidence from commercial insurance data

PLOS ONE

Dear Dr. Stockbridge,

Thank you for submitting your manuscript to PLOS ONE. After careful consideration, we feel that it has merit but does not fully meet PLOS ONE’s publication criteria as it currently stands. Therefore, we invite you to submit a revised version of the manuscript that addresses the points raised during the review process.

Please submit your revised manuscript . If you will need more time than this to complete your revisions, please reply to this message or contact the journal office at plosone@plos.org. Please include the following items when submitting your revised manuscript:

We look forward to receiving your revised manuscript.

Kind regards,

Frederick Quinn

Academic Editor

PLOS ONE

Journal Requirements:

2.We note that you have indicated that data from this study are available upon request. PLOS only allows data to be available upon request if there are legal or ethical restrictions on sharing data publicly. For information on unacceptable data access restrictions, please see http://journals.plos.org/plosone/s/data-availability#loc-unacceptable-data-access-restrictions.

Reviewers' comments:

Reviewer's Responses to Questions

**Comments to the Author**

1. Is the manuscript technically sound, and do the data support the conclusions?

Reviewer #1: Yes

Reviewer #2: Yes

2. Has the statistical analysis been performed appropriately and rigorously? 

Reviewer #1: Yes

Reviewer #2: Yes

3. Have the authors made all data underlying the findings in their manuscript fully available?

Reviewer #1: Yes

Reviewer #2: Yes

4. Is the manuscript presented in an intelligible fashion and written in standard English?

Reviewer #1: Yes

Reviewer #2: Yes

5. Review Comments to the Author

Reviewer #1: I was confused as to whether the patients were receiving improper medical care, as opposed to the type of test being associated with LTBI treatment completion. To help fix this, please change 2 minor items below:

1.Please pull the EHR data within OPTUM to the claims data in the manuscript. Because the majority of the patients only had 1 TST, LTBI is primarily in foreign-born persons, and foreign born persons are more likely to get BCG, it is unclear if persons not completing the study were BCG TST positive and had a faulty diagnosis from their health care provider. If so, one could assume that those individuals would not (and should not) complete LTBI therapy. The linked EHR data would provide more clarity on whether the results from this study are possibly due to poor clinical care. I understand this will limit you to about 25% of the total number of patients, but it would reassure your conclusions.

2. Also, I was confused as to why persons who received the TST who were less than 5 or had repeated TST testing were not removed from the dataset. Those individuals where provided improper diagnostic care from the onset. Could the authors please remove these individuals from their study?

Reviewer #2: 1. Most of the data is based on the source from United States. Should the author define this in the title?

2. P9, should be “And 19.5%”

3. In the Results section, the authors had provided some tables which contain a lot of data. However, it lacks description of results. For example, the Table 3-5, the authors should at least briefly describe the content, instead of only saying something was shown in this table.

4. For clarity, the authors should consider combining Table 4 & 5. It’s pretty redundant as the 2 tables shares pretty similar items.

5. There’re a lot of oral statement, please make revision accordingly. As a scientific manuscript, all sentences should be written strictly in scientific terms.

6. PLOS authors have the option to publish the peer review history of their article (what does this mean?). If published, this will include your full peer review and any attached files.

Reviewer #1: No

Reviewer #2: No

---

## [Author Response · Author response to Decision Letter 0]

27 Oct 2020

October 23, 2020

Dear Dr. Frederick Quinn and PLOS ONE peer reviewers,

We would like to offer our sincere thanks for your reviews of and feedback on our manuscript. We revised the manuscript in accordance with your guidance. The changes and our responses are described in point-by-point detail below.

Editor

***

Thank you for pointing out that we inconsistently applied PLOS ONE’s style requirements. These have been corrected, including file names.

2.We note that you have indicated that data from this study are available upon request. PLOS only allows data to be available upon request if there are legal or ethical restrictions on sharing data publicly … In your revised cover letter, please address my prompts.

***

Please see the cover letter for an explanation and an updated data availability statement.

Reviewer #1

Reviewer #1: I was confused as to whether the patients were receiving improper medical care, as opposed to the type of test being associated with LTBI treatment completion. To help fix this, please change 2 minor items below:

1.Please pull the EHR data within OPTUM to the claims data in the manuscript. Because the majority of the patients only had 1 TST, LTBI is primarily in foreign-born persons, and foreign born persons are more likely to get BCG, it is unclear if persons not completing the

study were BCG TST positive and had a faulty diagnosis from their health care provider. If so, one could assume that those individuals would not (and should not) complete LTBI therapy. The linked EHR data would provide more clarity on whether the results from this study are

possibly due to poor clinical care. I understand this will limit you to about 25% of the total number of patients, but it would reassure your conclusions.

***

Thank you for this suggestion. We agree that electronic medical record (EMR) data might provide additional insights into LTBI testing and treatment in patients receiving LTBI-related care in the private sector. We mention the possibilities inherent in EMR data in the last sentence of our discussion section. However, we do not have a license for the Optum EMR data; our license is limited to the claims data from the Clinformatics Data Mart. Consequently, we cannot incorporate these data into our analyses. Even so, claims data are an increasingly important window into the LTBI testing and treatment occurring in the private sector (see references #9, #14, #25, #60 and #61). 

Given current and previous US Centers for Disease Control and Prevention (CDC) practice guidelines, the widespread TST use observed in our study is not suggestive of improper medical care. Although IGRA testing is preferable, TST use is not contraindicated for persons with BCG vaccination. Specifically, the CDC guidelines in place during the period of study indicate that “an IGRA is preferred but a TST is acceptable” when testing BCG-vaccinated persons (see https://www.cdc.gov/mmwr/preview/mmwrhtml/rr5905a1.htm?s_cid=rr5905a1_e), and more recent guidelines recommend IGRAs for this population but indicate that “a TST is an acceptable alternative” (see https://www.thoracic.org/statements/resources/tb-opi/diagnosis-of-tuberculosis-in-adults-and-children.PDF). Our study aligns with past studies of LTBI testing; while IGRA use is increasing, TSTs have been the most common testing method in the private sector (see Owusu-Edusei 2017, reference #14).

Additionally, it is unlikely that the inappropriate initiation of treatment is driving the observed association between TSTs and treatment non-completion. Prior to initiating LTBI treatment, active TB must be ruled out. This process, which includes a chest radiograph and personal history, provides ample opportunity for providers to identify prior BCG vaccination and assess the likelihood of false positives (see our discussion section for details of this process). We found no literature suggesting that it is common for improper medical care to result in the initiation of LTBI after inaccurate LTBI diagnoses. In fact, prior studies suggest that persons testing positive for Mycobacterium tuberculosis are at high risk of not initiating treatment when it is indicated – it is likely that non-initiation of treatment when indicated is a much greater issue than initiation of treatment when it is not indicated. Of persons in the general population, only slightly over half of those tested and identified as eligible for LTBI treatment actually start treatment (see Alsdurf et al. 2016, reference #6). 

That said, we are unable to rule out the possibility that one or more persons in our study had a false-positive test and subsequently discontinued treatment once the erroneous results were identified. Thus, we have added this limitation to the second to the last paragraph of the discussion section. Text has been added which states, “Data limitations left us unable to determine if LTBI treatment was initiated after a false-positive test or whether there are associations between false-positive test results and treatment completion. Still, LTBI is seldom diagnosed or treatment started based solely on test result, and while it is plausible this gap could affect the magnitude of our findings it is unlikely to challenge our conclusions.”

2. Also, I was confused as to why persons who received the TST who were less than 5 or had repeated TST testing were not removed from the dataset. Those individuals where provided improper diagnostic care from the onset. Could the authors please remove these individuals from their study?

***

Thank you for this question. There is likely more variation for TB risk management practices in the private healthcare sector (which this data represents) than in the more common public health department setting, and it is of interest to capture and reflect the broadest range of practice these data allow. As you suggest, it is plausible this approach may at times capture improper diagnostic care. We are uncertain that this is an example of that, however--CDC guidelines from 2000 and 2016 both recommend TST testing for children less than 5 years of age (see guidelines from the year 2000 -- https://www.cdc.gov/mmwr/preview/mmwrhtml/rr4906a1.htm -- as well as more recent guidelines from 2016 -- https://www.thoracic.org/statements/resources/tb-opi/diagnosis-of-tuberculosis-in-adults-and-children.PDF). 

Additionally, CDC guidance indicates that repeated TST testing is warranted in certain situations. Guidelines indicate that persons at low risk for Mycobacterium tuberculosis infection should not be tested; however, legal or credentialing bodies may require such testing. When LTBI testing is conducted on low-risk persons and the test is positive, a second test is appropriate. Guidelines state that “The confirmatory test may be either an IGRA or a TST. When such testing is performed, the person is considered infected only if both tests are positive.” (see page 13 of https://www.thoracic.org/statements/resources/tb-opi/diagnosis-of-tuberculosis-in-adults-and-children.PDF). Other CDC guidelines have called for multiple TSTs to be conducted in healthcare workers (i.e., “two-step testing,” see https://www.cdc.gov/mmwr/pdf/rr/rr5417.pdf and https://www.cdc.gov/tb/publications/ltbi/pdf/LTBIbooklet508.pdf ). 

Given the CDC guidance we have retained these individuals in our analyses. To clarify the appropriateness of multiple testing, we have added a sentence to the fourth paragraph of our introduction section. It describes how “… clinical practice guidelines state that when persons at low risk for LTBI are tested and the results are positive, a second test is appropriate.” 

Reviewer #2

Reviewer #2: 1. Most of the data is based on the source from United States. Should the author define this in the title?

***

Good idea – thank you. The title has been revised to read: Interferon gamma release assay tests are associated with persistence and completion of latent tuberculosis infection treatment in the United States: evidence from commercial insurance data 

2. P9, should be “And 19.5%”

***

Thank you for pointing out this issue. We removed the unnecessary period before “and” so that the capitalization is no longer required.

3. In the Results section, the authors had provided some tables which contain a lot of data. However, it lacks description of results. For example, the Table 3-5, the authors should at least briefly describe the content, instead of only saying something was shown in this table.

***

Thank you for pointing out that it appears that we do not describe the content of the tables. We have rearranged the content – originally we introduced tables 3-5 with a brief paragraph, presented the tables, and then described them after presentation. We have rearranged some of the content that originally followed the tables. You will see now that there is information describing the content of the tables, with the detailed information about the unadjusted primary findings preceding tables 3-4 and the remaining information following the tables.

4. For clarity, the authors should consider combining Table 4 & 5. It’s pretty redundant as the 2 tables shares pretty similar items.

***

Thank you for this suggestion. The two tables are combined and the combined table is now Table 4.

5. There’re a lot of oral statement, please make revision accordingly. As a scientific manuscript, all sentences should be written strictly in scientific terms.

***

Thank you for this feedback. We have revised the manuscript in accordance with this guidance. Specifically, in the discussion section the sentence “Preserving the value of this investment as that patient ‘comes down the home stretch’ is disproportionately impactful” was changed to “Preserving the value of this investment as that patient nears the end of the lengthy LTBI care continuum is disproportionately impactful.”

Once again, thank you for the positive and helpful feedback. Please contact us with any questions or concerns about how we addressed the guidance that we received. 

Sincerely, 

Erica L. Stockbridge, PhD 

Corresponding Author

---

## [Decision Letter · Decision Letter 1]

16 Nov 2020

Interferon gamma release assay tests are associated with persistence and completion of latent tuberculosis infection treatment in the United States: Evidence from commercial insurance data

PONE-D-20-22812R1

Dear Dr. Stockbridge,

We’re pleased to inform you that your manuscript has been judged scientifically suitable for publication and will be formally accepted for publication once it meets all outstanding technical requirements.

Kind regards,

Frederick Quinn

Academic Editor

PLOS ONE

Additional Editor Comments (optional):

Reviewers' comments:

Reviewer's Responses to Questions

**Comments to the Author**

1. If the authors have adequately addressed your comments raised in a previous round of review and you feel that this manuscript is now acceptable for publication, you may indicate that here to bypass the “Comments to the Author” section, enter your conflict of interest statement in the “Confidential to Editor” section, and submit your "Accept" recommendation.

Reviewer #2: (No Response)

2. Is the manuscript technically sound, and do the data support the conclusions?

Reviewer #2: Yes

3. Has the statistical analysis been performed appropriately and rigorously? 

Reviewer #2: Yes

4. Have the authors made all data underlying the findings in their manuscript fully available?

Reviewer #2: Yes

5. Is the manuscript presented in an intelligible fashion and written in standard English?

Reviewer #2: Yes

6. Review Comments to the Author

Reviewer #2: (No Response)

7. PLOS authors have the option to publish the peer review history of their article (what does this mean?). If published, this will include your full peer review and any attached files.

Reviewer #2: No

---

## [Editor Report · Acceptance letter]

24 Nov 2020

PONE-D-20-22812R1 

Interferon gamma release assay tests are associated with persistence and completion of latent tuberculosis infection treatment in the United States: Evidence from commercial insurance data 

Dear Dr. Stockbridge:

I'm pleased to inform you that your manuscript has been deemed suitable for publication in PLOS ONE. Congratulations! Your manuscript is now with our production department. 

Kind regards, 

on behalf of

Dr. Frederick Quinn 

Academic Editor

PLOS ONE